# Potential of psychrotolerant rhizobacteria for the growth promotion of wheat (*Triticum aestivum* L.)

Muhammad Abdullah[1], Mohsin Tariq[1], Syeda Tahseen Zahra[1], Azka Ahmad[2], Marriam Zafar[1] and Shad Ali[1]

[1] Bioinformatics and Biotechnology, Government College University Faisalabad, Faisalabad, Punjab, Pakistan
[2] Center of Excellence in Molecular Biology (CEMB), University of the Punjab, Lahore, Punjab, Pakistan

## ABSTRACT

Wheat is the second most important staple crop grown and consumed worldwide. Temperature fluctuations especially the cold stress during the winter season reduces wheat growth and grain yield. Psychrotolerant plant growth-promoting rhizobacteria (PGPR) may improve plant stress-tolerance in addition to serve as biofertilizer. The present study aimed to isolate and identify PGPR, with the potential to tolerate cold stress for subsequent use in supporting wheat growth under cold stress. Ten psychrotolerant bacteria were isolated from the wheat rhizosphere at 4 °C and tested for their ability to grow at wide range of temperature ranging from −8 °C to 36 °C and multiple plant beneficial traits. All bacteria were able to grow at 4 °C to 32 °C temperature range and solubilized phosphorus except WR23 at 4 °C, whereas all the bacteria solubilized phosphorus at 28 °C. Seven bacteria produced indole-3-acetic acid at 4 °C, whereas all produced indole-3-acetic acid at 28 °C. Seven bacteria showed the ability to fix nitrogen at 4 °C, while all the bacteria fixed nitrogen at 28 °C. Only one bacterium showed the potential to produce cellulase at 4 °C, whereas four bacteria showed the potential to produce cellulase at 28 °C. Seven bacteria produced pectinase at 4 °C, while one bacterium produced pectinase at 28 °C. Only one bacterium solubilized the zinc at 4 °C, whereas six bacteria solubilized the zinc at 28 °C using ZnO as the primary zinc source. Five bacteria solubilized the zinc at 4 °C, while seven bacteria solubilized the zinc at 28 °C using $ZnCO_3$ as the primary zinc source. All the bacteria produced biofilm at 4 °C and 28 °C. In general, we noticed behavior of higher production of plant growth-promoting substances at 28 °C, except pectinase assay. Overall, *in vitro* testing confirms that microbes perform their inherent properties efficiently at optimum temperatures rather than the low temperatures due to high metabolic rate. Five potential rhizobacteria were selected based on the *in vitro* testing and evaluated for plant growth-promoting potential on wheat under controlled conditions. WR22 and WR24 significantly improved wheat growth, specifically increasing plant dry weight by 42% and 58%, respectively. 16S *rRNA* sequence analysis of WR22 showed 99.78% similarity with *Cupriavidus campinensis* and WR24 showed 99.9% similarity with *Enterobacter ludwigii*. This is the first report highlighting the association of *C. campinensis* and *E. ludwigii* with wheat rhizosphere. These bacteria can serve as potential candidates for biofertilizer to mitigate the chilling effect and improve wheat production after field-testing.

Corresponding author
Mohsin Tariq,
mohsintariq@gcuf.edu.pk

## INTRODUCTION

Wheat is the second most important crop after rice, contributing 35% of total global grain production. Global wheat demand is estimated to increase up to 70% by 2050 to feed the ever-increasing human population (*Vitale, Adam & Vitale, 2020*). Pakistan annually produces 26.4 million tons of wheat, which contributes 1.8% to the GDP (*Economic Survey of Pakistan, 2021-22*). Improving crop productivity has become a very important goal in satisfying food demand (*Vitale, Vitale & Epplin, 2019*). Climate change is the major limiting factor in wheat productivity and poses a constant threat to food security (*Hassan et al., 2021*). Climate change intensifies abiotic stresses such as heat and cold, which reduces crop production (*Ding, Shi & Yang, 2019*; *Hussain et al., 2018*). Cold stress is a major abiotic stress, in which plant experiences lower temperatures ranging from 0 °C to 15 °C (*Solanke & Sharma, 2008*; *Miura & Furumoto, 2013*). During cold stress, the plant produces reactive oxygen species (ROS) that damage the macromolecules and ultimately disrupt cellular homeostasis (*Suzuki & Mittler, 2006*; *Ruelland et al., 2009*; *Gill & Tuteja, 2010*). Wheat is cultivated during the winter season in temperate regions and 85% of its area endures cold stress (*Hassan et al., 2021*).

Different methods have been used to deal with cold stress, such as sprinkler irrigation, wind machines, and heaters (*Poling, 2008*). Chemical fertilizers are usually used to enhance crop production and mitigate plant stresses, but these fertilizers jeopardize environmental sustainability (*Srivastav, 2020*). Biofertilizers are based upon plant-beneficial bacteria that just like the chemical fertilizers, enhance soil fertility and nutrient availability for plants but do not harm the environment (*Adesemoye, Torbert & Kloepper, 2009*; *Morcillo & Manzanera, 2021*). These microbes in the rhizosphere facilitate plants to cope with biotic and abiotic stresses (*Dai et al., 2020*) by upregulating the expression of stress-responsive genes (*Berendsen, Pieterse & Bakker, 2012*; *Bakker et al., 2018*). Rhizospheric microorganisms such as *Pseudomonas*, *Bradyrhizobium japonicum*, and *Bacillus* are known to display abiotic stress tolerance and rescue stressed plants (*Seneviratne et al., 2017*; *Ma et al., 2020*; *Lalay, Ullah & Ahmed, 2022*). The wheat rhizosphere is a niche for many bacteria, including *Paenibacillus, Bacillus subtilis, Bacillus thuringiensis, Arthrobacter nicotianae, Micrococcus luteus, Bacillus amyloliquefaciens, Pantoea, Pseudomonas, Methylobacterium, Agrobacterium* (*Bhattacharyya & Jha, 2012*; *Verma & Suman, 2018*; *Mahapatra et al., 2020*; *Wang et al., 2021*; *Zahra et al., 2023*). Microbial diversity and plant growth-promoting traits of microbes are severely affected by adverse environmental conditions such as cold stress (*Zeng, Wu & Wen, 2016*; *Vega-Celedon et al., 2021*). Low temperature cause ice crystallization in the microbial membrane, affecting the activity of essential enzymes reducing their activities to 2–4 folds for every 10 °C decrease in the temperature (*Rizvi et al., 2021*). Moreover, plant growth-promoting characteristics such as phosphate solubilization,

and IAA production also get influenced by low temperatures, affecting the plant–microbe symbiosis which further decreases the plant growth rate (*Meena et al., 2015*).

Psychrotolerant bacteria, on the other hand, are the bacterial communities that thrive in low-temperature conditions (*Jin, Wang & Zhao, 2022*). These bacteria prevent ice crystal formation in their membranes by synthesizing anti-freezing proteins, thereby regulating downstream processes (*Cid et al., 2017*). In addition, psychrotolerant bacteria can improve plant growth under cold stress conditions through nitrogen fixation, phosphate solubilization, and phytohormone production even at 4 °C (*Mishra et al., 2011*). Hence, the use of plant-beneficial psychrotolerant bacteria is promiscuous for optimizing agricultural production while mitigating the negative effects of cold stress (*Adhikari et al., 2021*). Psychrotolerant bacteria belonging to the genera *Pseudomonas, Burkholderia, Bradyrhizobium, Azospirrilum, Bacillus, and Raoultella* demonstrate the potential to increase plant growth under cold stress (*Zhang et al., 2003*; *Ait Barka, Nowak & Clément, 2006*; *Mishra et al., 2009*; *Turan et al., 2013*).

Our hypothesis was that psychrotolerant bacteria may play a significant role in developing a cold-tolerant biofertilizer for wheat. In this study, psychrotolerant bacteria were isolated from wheat roots, identified phylogenetically, and evaluated for their potential to promote plant growth and mitigate cold stress.

## MATERIALS & METHODS

### Sample collection and isolation of psychrotolerant bacteria

One-month-old wheat, cultivar Akbar-19, plants were collected in February 2022 from a field located in Rawalakot (GPS coordinates at 25°4′26.7456″N and 68°45′47.4912″E), Azad Jammu and Kashmir (AJK). Samples were transported to the Plant Biotechnology Lab of Government College University Faisalabad, Pakistan, and stored at 4 °C. Isolation of rhizobacteria was performed under aseptic conditions according to *Tsegaye et al. (2019)* with some modifications. Roots were washed with sterilized water to detach the adhering soil. Roots (1 g) were separated from the plant and transferred to the test tube containing nine mL of saline solution (0.85% NaCl). Tubes were vortexed to remove bacteria from the root surface and serially diluted up to $10^{-5}$. An aliquot of 100 μL from each dilution was spread on the King's B media (Peptone 20 g $L^{-1}$, $K_2HPO_4$ 1.5 g $L^{-1}$, $MgSO_4.7H_2O$ 1.5 g $L^{-1}$, glycerol 10 g $L^{-1}$, agar 15 g $L^{-1}$) and incubated at 4 ± 2 °C for 48 h (*King, Ward & Raney, 1954*). Bacterial colonies showing different morphologies were selected and purified by sub-culturing on fresh King's B media plates (*Mishra et al., 2009*).

### Bacterial temperature tolerance assay

The bacterial isolates were tested for their ability to grow at different temperatures. Isolates were streaked on to King's B media plates and incubated at each of the temperatures (−8 °C, −4 °C, 0 °C, 4 °C, 8 °C, 12 °C, 16 °C, 20 °C, 24 °C, 28 °C, 32 °C and 36 °C) for 48 h. Growth of bacteria was recorded as arbitrary values high (+++), moderate (++), low (+) or negative (-) by visual observation (*Subramanian et al., 2016*; *Hassan et al., 2021*).

## Screening of plant growth-promoting characteristics

### Phosphate solubilization

The phosphate solubilization ability of bacterial isolates was determined by inoculating a single colony of each bacterial isolate on Pikovskaya's agar plate containing tricalcium phosphate (*Pikovskaya, 1948*) and incubated at 4 ± 2 °C and 28 ± 2 °C for 7 days. After incubation, the plates were observed for clear zone formation around the colonies, indicating phosphate solubilization, and the phosphate solubilization index (PSI) was calculated according to the below-mentioned formula (*Nacoon et al., 2020*).

PSI = diameter of halo zone/diameter of colony

### Indole-3-acetic acid (IAA) production

Indole-3-acetic acid (IAA) production was estimated by Salkowski's calorimetric method with some modifications. Bacterial cultures were grown in King's B broth, supplemented with tryptophan, and incubated at 4 ± 2 °C and 28 ± 2 °C for 3 days. After incubation, the bacterial cultures were centrifuged at 10,000 rpm for 10 min to remove bacterial cells, and the supernatant was collected. A total of 1 mL of the supernatant was mixed with 2 mL of Salkowski's reagent (2 mL 0.5 M ferric chloride and 100 mL 35% (w/w) perchloric acid) and incubated at room temperature for 25 min in the dark. Pink coloration indicated the IAA production. The absorbance was recorded at 530 nm using a spectrophotometer and the amount of IAA was quantified by comparing it with the standard curve (*Myo et al., 2019*; *Hyder et al., 2020*).

### Nitrogen fixation

The nitrogen-fixing ability of the isolates was determined by streaking the single colony on nitrogen-free media (mannitol 20 g $L^{-1}$, $K_2HPO_4$ 0.2 g $L^{-1}$, NaCl 0.2 g $L^{-1}$, $MgSO_4$ 0.2 g $L^{-1}$, $K_2SO_4$ 0.1 g $L^{-1}$, $CaCO_3$ 5.0 g $L^{-1}$, agar 20 g $L^{-1}$) and incubated at 4 ± 2 °C and 28 ± 2 °C for 3 days. The growth of all bacteria was recorded as arbitrary values strong (+++), moderate (++), weak (+), or negative indicating their ability to fix nitrogen (*Mirza & Rodrigues, 2012*).

### Zinc solubilization

The zinc solubilization ability of the bacterial isolates was checked by inoculating the individual colonies on tris-minimal media supplemented with ZnO and $ZnCO_3$ separately as a zinc source. Plates were incubated at 4 ± 2 °C and 28 ± 2 °C for 7 days in the dark. The formation of a halo zone around the bacterial colonies indicates the zinc solubilization ability of the bacterial isolates. The zinc solubilization efficiency (ZSE) was calculated according to the below-mentioned formula (*Rezaeiniko, Enayatizamir & Norouzi Masir, 2022*).

ZSE = (diameter of halo zone/diameter of the colony) ×100.

### Cellulase production assay

Cellulase activity of bacterial isolates was measured by using carboxymethylcellulose (10 g $L^{-1}$) agar plates ($MgSO_4$ 0.25 g $L^{-1}$, $K_2HPO_4$ 10 g $L^{-1}$, yeast extract 5 g $L^{-1}$, peptone 5 g $L^{-1}$, gelatin 2 g $L^{-1}$, cellulose 10 g $L^{-1}$, agar 20 g $L^{-1}$). Bacterial cultures were spot inoculated on the center of the carboxymethylcellulose agar plates and incubated at 4 ± 2 °C

and 28 ± 2 °C for 7 days (*Islam & Roy, 2018*). After incubation, plates were stained with Congo red dye (0.2%), incubated at room temperature for 15 min, and washed with 1 M NaCl solution. A clear zone around the colonies was observed and the cellulase activity index (CAI) of bacterial isolates was calculated according to the below-mentioned formula (*Suárez-Moreno et al., 2019*).

CAI = diameter of halo zone/diameter of the colony

### Pectinase production assay

The pectinase activity of the bacterial isolates was assessed by spotting bacterial colonies on the center of pectin (10 g $L^{-1}$) agar plates (KCl 1 g $L^{-1}$, $NaNO_3$ 1 g $L^{-1}$, $MgSO_4$ 0.5 g $L^{-1}$, $K_2HPO_4$ 1 g $L^{-1}$, pectin 10 g $L^{-1}$, yeast extract 0.5 g $L^{-1}$, agar 20 g $L^{-1}$) and incubated at 4 ± 2 °C and 28 ± 2 °C for 7 days (*Devi et al., 2022*). After incubation, the plates were stained with 1% iodine solution for 15 min and washed with sterilized water. A clear zone around the colonies was observed and the pectinase activity index (PAI) of bacterial isolates was calculated according to the below-mentioned formula (*Tsegaye et al., 2019*).

PAI = diameter of halo zone/diameter of the colony

### Biofilm formation assay

Biofilm formation was tested using a microtiter plate assay (*Kasim et al., 2016*). Bacterial cultures were grown in King's B broth medium up to 2 optical density at 600 nm ($OD_{600\ nm}$) and centrifuged at 6,000 rpm for 2 min. The supernatant was discarded and the pellet carrying bacterial cells was washed with sterile water. Bacterial cells were resuspended in King's B broth and diluted to an $OD_{600\ nm}$ of 0.2. An aliquot of 150 µL of each bacterial isolate was added to the 96-well polyvinyl chloride (PVC) plate in six replications and incubated at 28 ± 2 °C for 48 h. Bacterial cultures were removed and wells were washed with sterile water. Wells were stained with 150 µL of crystal violet (0.001%) for 15 min. The stain was removed with a pipette, followed by washing with sterile water and air drying. Absorbed dye in the wells was solubilized by adding 150 µL of 95% ethanol. Absorbance was measured at $OD_{570\ nm}$ in a microtiter plate reader, which presents the quantity of biofilm formation (*Jhuma et al., 2021*).

### Plant-inoculation experiment under controlled conditions

Controlled-conditions experiment was performed on the wheat cultivar Akbar-19 to test six treatments (WR15, WR16, WR18, WR22, WR24, and water as a control) with three replicates in a completely randomized design (CRD). Seeds were surface sterilized with 70% ethanol for 1 min and sodium hypochlorite (3%) for 30 s, followed by washing with sterile water. Sterilized seeds were placed on wet filter paper in a petri plate and incubated at room temperature for 48 h in the dark. Uniform-sized germinated seeds were transferred to the pots containing sterilized soil. Bacterial cultures were grown in King's B media and centrifuged at 6,000 rpm for 10 min. The bacterial pellets of each culture was resuspended in sterile water to obtain an $OD_{600nm}$ of 0.5 (*Mishra et al., 2009*). After one week of germination, 100 µL of inoculum from each bacterial treatment was applied to the root of the plant and placed in a growth chamber at 15 ± 2 °C during the day and 10 ± 2 °C at night. Plants were watered with sterile water and Hoagland's solution on alternative days.

After six weeks of inoculation, agronomical parameters such as shoot fresh weight, root fresh weight, shoot dry weight, root dry weight, root length, and shoot length of the plants were recorded and statistically analyzed (*Tounsi-Hammami et al., 2022*). Statistical analysis was done using ANOVA and LSD ($p$ value = 0.05) using CoStat software (*Cardinali & Nason, 2013*).

### Phylogenetic analysis of psychrotolerant bacteria

Phylogenetic identification of potential bacterial isolates was performed by 16S rRNA gene amplification using universal primers fD1 (5′-AGAGTTTGATCCTGGCTCAG-3′) and rD1 (5′–AAGGAGGTGATCCAGCC-3′) (*Weisburg et al., 1991*). A total of 50 μL PCR mixture contained Taq polymerase buffer (10X) 5 μL, dNTPs (2 mM) 5 μL, primers (10 mM) 2 μL each, $MgCl_2$ (25 mM) 4 μL, water 23. 4 μL, Taq polymerase (5 U $\mu L^{-1}$) 0.6 μL and template DNA 4 μL). The PCR mixture was placed in the thermocycler under the thermal profile of initial denaturation at 95 °C for 5 min followed by 30 cycles of denaturation at 95 °C for 1 min, annealing at 55 °C for 50 s, extension at 72 °C for 1 min 40 s and final extension at 72 °C for 5 min. Amplification of the 16S rRNA gene was confirmed on 1% agarose gel. Amplified PCR products were purified using the Thermo Scientific GeneJet PCR Purification Kit (cat # K0701) and Sanger sequenced using services of Macrogen, Korea. After sequencing, contigs were formed and sequences were compared using the NCBI BLAST tool (*Altschul et al., 1990*). The phylogenetic tree was constructed with authenticated sequences retrieved from the database by the maximum likelihood method using Mega 11 software with 1,000 bootstrap values and a pairwise identity chart was also constructed using Sequence Demarcation Tool (SDT) v.1.2 (*Kumar, Stecher & Tamura, 2016*; *Noori et al., 2021*).

## RESULTS

### Isolation of psychrotolerant bacteria

Different bacterial colonies with varying colors, shapes, edges, and sizes were obtained on the King's B media agar plates. A total of 10 bacteria were selected based on colony morphology. The morphological characteristics of colonies were highly variable, as shown in Table 1.

### Bacterial temperature tolerance assay

All the bacteria were tested for their temperature tolerance ability by assessing growth at different temperatures ranging from −8 to 36 °C. WR15, WR17, WR22, and WR24 were able to grow at −4 °C. WR16, WR19, WR20, WR22 and WR24 showed growth upto 36 °C. All the bacterial isolates showed growth from 4 °C to 32 °C. The bacterial growth at different temperatures is shown in Fig. 1. WR22 and WR24 showed growth at all temperatures, ranging from −4 to 36 °C.

### Plant growth-promoting characteristics

#### *Phosphate solubilization*

All the bacteria showed phosphate solubilization ability at 4 °C except WR23 with the PSI range of 1.11–1.91, where isolate WR24 showed maximum P-solubilization potential. At

**Table 1 Colony morphology of wheat rhizobacteria.**

| Name | Color | Appearance | Edges | Size |
|---|---|---|---|---|
| WR15 | White | Gummy | Round | Large |
| WR16 | Pale-yellow | Non-gummy | round | Small |
| WR17 | Red | Non-gummy | Irregular | Very small |
| WR18 | Yellow | Flat | Irregular | Very small |
| WR19 | Off-white | Flat | Irregular | Medium |
| WR20 | White | Gummy | Round | Small |
| WR21 | Off-white | Non-gummy | Round | Medium |
| WR22 | Pale Yellow | Non-gummy | Irregular | Large |
| WR23 | White | Non-gummy | Round | Small |
| WR24 | Off-white | Gummy | Round | Large |

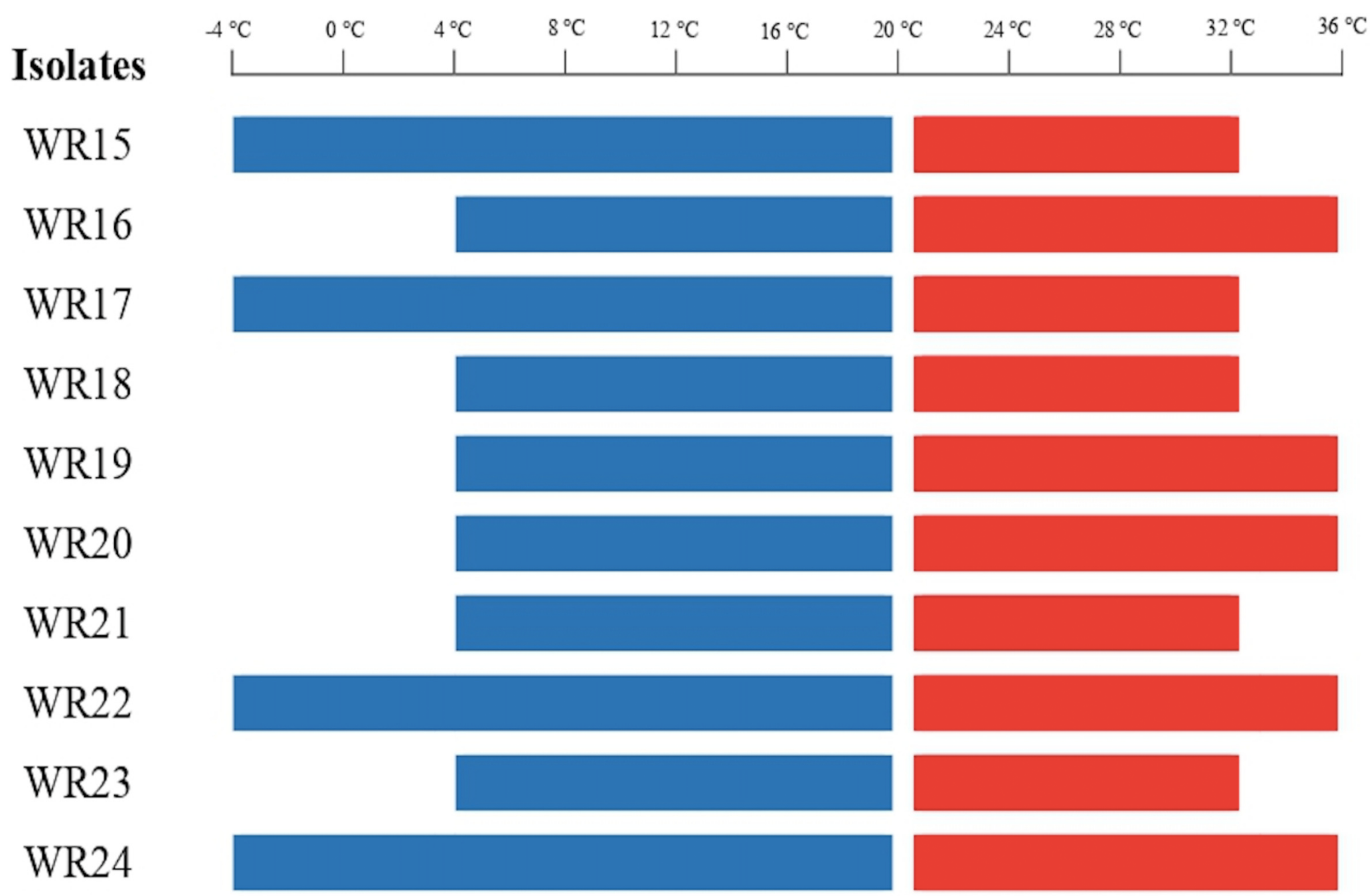

**Figure 1 Bacterial growth at different temperatures ranging from −4 °C to 36 °C.** Red lines present above 20 °C and blue lines present below 20 °C.

28 °C, all bacterial isolates showed P-solubilization showing PSI index of 1.06–2.6 where isolate WR19 showed maximum P-solubilization potential. The P-solubilization ability of the bacteria at 4 °C and 28 °C is shown in Figs. 2 and 3.

### *Indole-3-acetic acid production*

IAA production ability of bacterial isolates WR15, WR17, WR18, WR19, WR20, WR22 and WR24 at 4 °C ranged from 0–4.28 μg $mL^{-1}$, where isolate R24 showed the highest IAA production. At 28 °C, all the bacteria showed IAA production with the range of

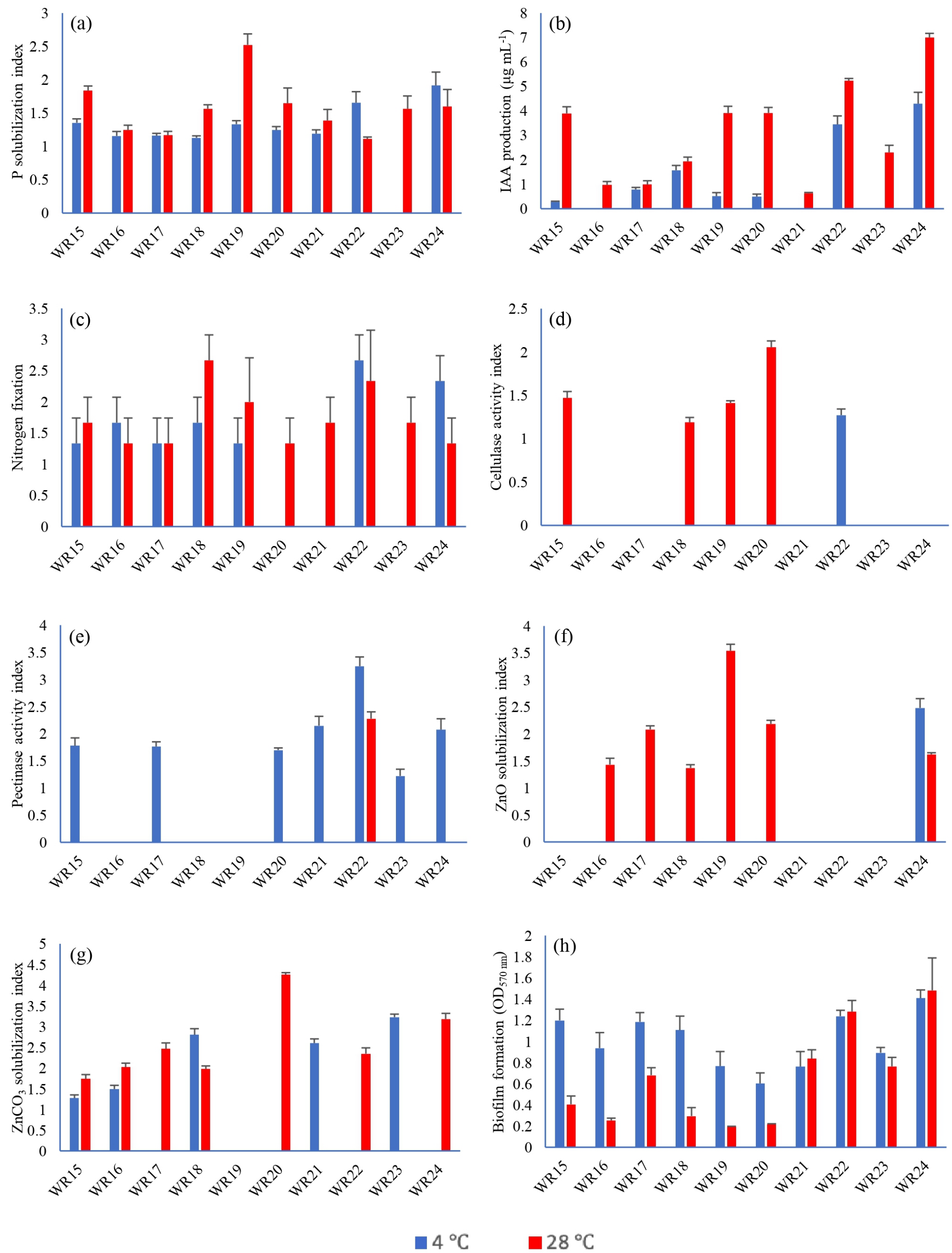

**Figure 2 Biochemical characteristics of rhizospheric bacteria at 4 °C and 28 °C.** Tested bacteria showed phosphate solubilization (A), indole-3-acitic acid production (B), nitrogen fixation (C), cellulase production (D), pectinase production (E), zinc oxide solubilization (F), zinc carbonate solubilization (G) and biofilm formation (H) [$n = 3$].

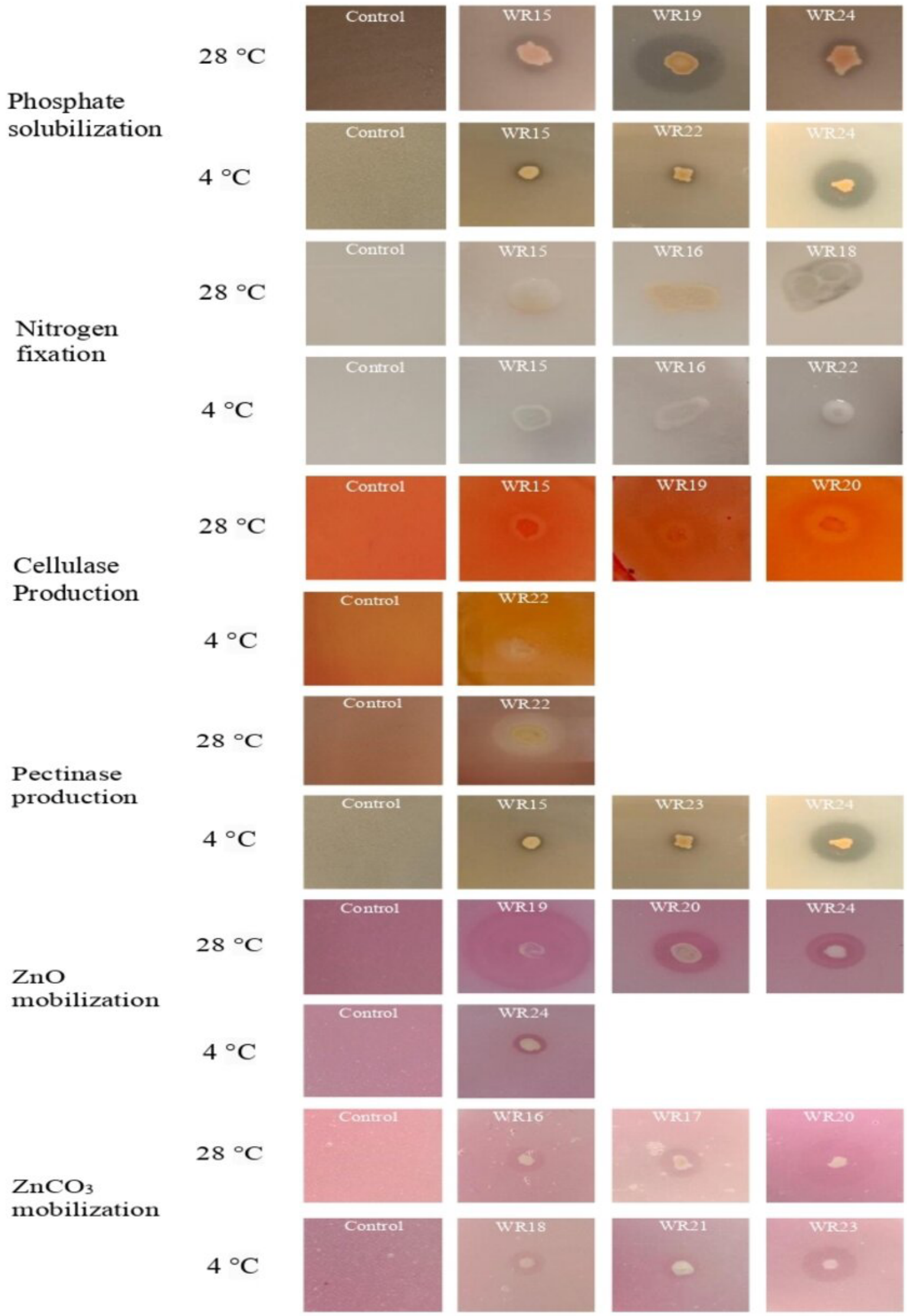

**Figure 3 Biochemical characteristics of potential rhizospheric bacteria at 4 °C and 28 °C.** The most potential isolates for each test are presented.

0.62–7 μg $mL^{-1}$ where, WR 24 showed the highest IAA production. The IAA production ability of all the bacterial isolates is shown in Fig. 2.

### *Nitrogen fixation*

Bacterial isolates, WR15, WR16, WR17, WR18, WR19, WR22 and WR24, showed the ability to fix nitrogen at 4 °C and maximum nitrogen fixation was shown by WR22. All the bacteria showed nitrogen fixation at 28 °C and maximum nitrogen fixation (+++) was demonstrated by WR18. The nitrogen fixation ability of the bacterial isolates is shown in Figs. 2 and 3.

### *Zinc oxide and zinc carbonate solubilization*

In the zinc oxide solubilization assay, only WR24 showed halo zone formation with an index of 2.5 at 4 °C. WR16, WR17, WR18, WR19, WR20, and WR24 showed zinc oxide solubilization at 28 °C ranging 1.43–3.5 index, in which WR19 showed maximum zinc oxide solubilization at 28 °C by producing 3.5 index. In zinc carbonate solubilization assay, WR15, WR16, WR18, WR21, and WR23 showed Zn-solubilization at 4 °C with an index ranging from 1.27–3.2 while the isolates WR15, WR16, WR17, WR18, WR20, WR22 and WR24 showed Zn-carbonate solubilization at 28 °C with ZSI 1.78–4.3, where WR20 showed maximum zinc solubilization. The zinc oxide and zinc carbonate solubilization abilities of the bacteria are shown in Figs. 2 and 3.

### *Cellulase production*

Only one bacterial isolate, WR22, showed cellulase production ability at 4 °C with an activity index of 1.3. At 28 °C, WR15, WR18, WR19, WR20 showed cellulase production with an activity index of 1.17–2.1. The cellulase activity index is presented in Figs. 2 and 3.

### *Pectinase production*

Pectinase production ability was detected in WR15, WR17, WR20, WR21, WR22, WR23 and WR24 at 4 °C with and index of 1.12–3.2. At 28 °C, only one bacterial isolate, WR22, showed pectinase production with an index of 2.3. The pectinase production is shown in Figs. 2 and 3.

### *Biofilm formation*

All the bacteria showed biofilm formation (Fig. 2) at 4 °C and 28 °C ranging from 0.76–1.37 OD at 4 °C 0.23–1.31 OD at 28 °C, where WR24 showed maximum biofilm formation ability.

## Plant-inoculation experiment under controlled conditions

The inoculated plants grown under controlled conditions were healthy. Inoculation with bacterial isolates WR22 and WR24 showed a significant increase in shoot length compared to non-inoculated control plants (Fig. 4). WR24-inoculated plants showed a significant increase in root length. All bacterial treatments showed a significant increase in shoot fresh weight compared to control. In the case of shoot dry weight, WR24 showed a significant increase compared to the control. WR15, WR16, WR22, and WR24 showed a significant increase in root fresh weight. Whereas WR15, WR22, and WR24 showed a significant

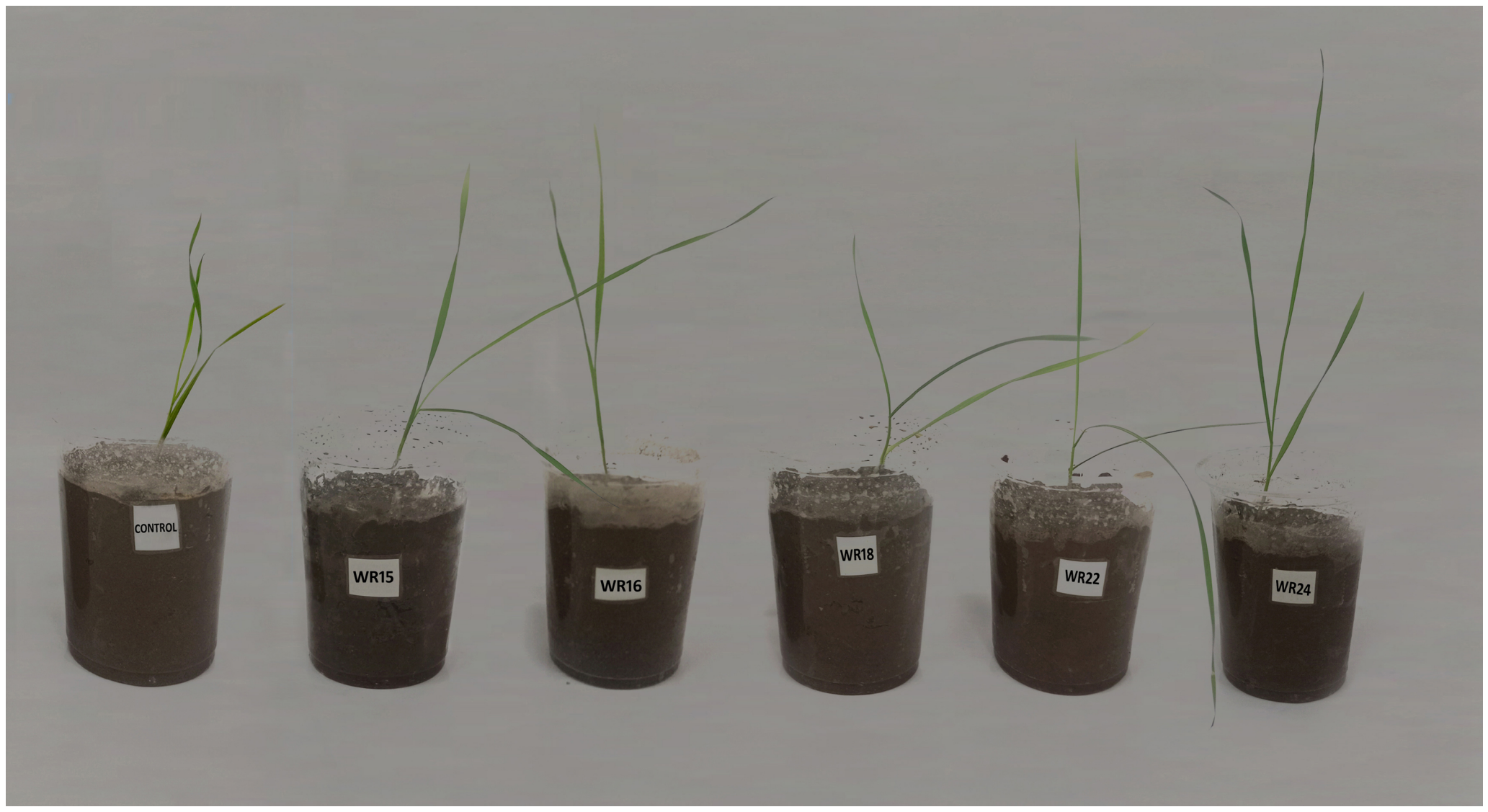

**Figure 4 Effect of potential bacteria on wheat growth under controlled conditions.** WR24 showed maximum potential to improve shoot length after 4 weeks of inoculation.

increase in root dry weight compared to control. Overall, WR24 showed maximum potential to increase shoot length, root length, shoot fresh weight, and shoot dry weight by 31%, 23%, 81%, and 58%, respectively, compared to the control. Inoculation with bacterial isolate WR22 showed the maximum increase in root fresh weight and root dry weight by 105% and 98%, respectively, compared to the control. WR18 showed negative effects on root fresh and dry weight by reducing 23.3% root fresh weight and 41% root dry weight as compared to the control. The effect of different treatments on agronomical parameters of wheat is shown in Table 2.

## Phylogenetic analysis

Amplified 1,500 bp product of 16S rRNA gene from bacteria WR22 and WR24 were sequenced commercially and sequence contigs were compared with database sequences using the NCBI BLAST tool. 16S rRNA gene from WR22 showed maximum similarity (99.34%) with *Cupriavidus campinensis* while WR24 with *Enterobacter ludwigii* (99.53%). Sequences were submitted to GenBank under the accession numbers OQ505159 for WR22 and OQ505160 for WR24. The phylogenetic tree was constructed using closely related authenticated sequences of *Cupriavidus* and *Enterobacter species*, where *Methanoregula boonei* was used as an outgroup. The tree was constructed based on the similarity of the 16S rRNA gene sequence of the species. Bacterial isolate WR22 was clustered in the neighborhood of *Cupriavidus campinensis* WS2, and bacterial isolate WR24 was clustered in the neighborhood of *Enterobacter ludwigii* EN-119T in the phylogenetic tree. WR24 was clustered in clade 1 which includes all the species of *Enterobacter*. WR22 was clustered in clade 2 which includes all the species of *Cupriavidus.* Each clade includes the species of the same genera because the species of the same genera have a maximum similarity of 16S

**Table 2** **Effect of potential bacteria on wheat growth under controlled conditions.**

| Isolates | Shoot length (cm) | Root length (cm) | Shoot fresh weight (mg) | Shoot dry weight (mg) | Root fresh weight (mg) | Root dry weight (mg) |
|---|---|---|---|---|---|---|
| Control | 16.3 ± 0.94[c] | 6.9 ± 0.55[b] | 82.2 ± 6.5[c] | 17 ± 1.28[b] | 52.3 ± 1.58[d] | 7.4 ± 0.58[c] |
| WR15 | 18.3 ± 1.09[bc] | 7.5 ± 0.5[ab] | 129.9 ± 6.71[ab] | 21.2 ± 1.68[ab] | 91.8 ± 8.3[b] | 12.9 ± 0.97[b] |
| WR16 | 18.4 ± 0.55[bc] | 8.4 ± 0.78[ab] | 111.4 ± 7.93[b] | 19.7 ± 3.5[ab] | 76.7 ± 6.9[c] | 9.9 ± 0.59[c] |
| WR18 | 16.1 ± 0.49[c] | 7.6 ± 0.53[ab] | 122.6 ± 8.7[b] | 22.3 ± 3.57[ab] | 40.1 ± 3.76[d] | 4.4 ± 0.45[d] |
| WR22 | 19.1 ± 1.18[ab] | 8.2 ± 0.48[ab] | 145.5 ± 7.2[a] | 24.2 ± 3.36[ab] | 107.6 ± 6.95[a] | 22.1 ± 1.85[a] |
| WR24 | 21.4 ± 0.89[a] | 8.5 ± 0.58[a] | 148.4 ± 7.06[a] | 26.9 ± 1.25[a] | 86.9 ± 3.49[bc] | 15.3 ± 1.35[b] |
| LSD | 2.36 | 1.48 | 19.48 | 7.34 | 14.95 | 2.87 |
| ANOVA significance | *** | ns | *** | ns | *** | *** |

**Notes.**

Each value represents mean ($n = 6$) ± standard error.

Values followed by the different letters in same column indicate significant difference and followed by same letters are not significantly different.

LSD and ANOVA tests were performed using CoStat software. LSD ($p$ value = 0.05) indicates least significant difference, (ns) indicates no significant, (*) indicates significant, (**) indicates moderately significant, (***) indicates highly significant.

rRNA gene sequence (Fig. 5). A pairwise identity chart was constructed using the SDT tool and the color-coded chart indicates the percentage identity between the species ranging from 80 to 100% (Fig. 6).

# DISCUSSION

Climate change is one of the most crucial problems worldwide and a major threat to food security (*FAO, 2020*). Cold stress exerts negative effects on the vegetative growth of wheat and ultimately results in yield losses affecting 85% of wheat cultivation annually (*Hassan et al., 2021*). Though cold tolerance can be induced in plants using genetic engineering tools, the process is quite time-consuming, tricky, and laborious. On the other hand, the use of plant growth-promoting rhizobacteria (PGPR) to rescue cold-affected plants is a quite simple, effective, and eco-friendly technology (*García et al., 2017*).

Numerous studies have demonstrated that psychrotolerant PGPR isolated from plants growing in cold regions has the potential to upregulate cold-tolerance mechanisms in plants such as wheat, grapevine, canola, tomato *etc*. (*Cheng, Park & Glick, 2007*; *Mishra et al., 2011*; *Verma et al., 2015*; *Subramanian et al., 2016*; *Singh & Khanna, 2017*). In this study, ten cold-tolerant bacteria were isolated from wheat rhizosphere. Isolation was carried out at 4 °C considering that bacteria present in the sample are adapted to grow at low temperatures. Our results are in agreement with previous studies showing that bacteria can tolerate different temperatures at 4 °C, 5 °C, 8 °C 15 °C, 16 °C, 20 °C, and 28 °C (*Mishra et al., 2011*; *Subramanian et al., 2016*; *Yarzábal et al., 2018*). Later, the bacteria were tested for their ability to grow at extremely low temperatures (−4 °C). Four bacteria were efficient enough to show growth even at −4 °C. Such extreme cold temperature conditions develop in many wheat cultivation areas of the world (*Frederiks et al., 2015*; *Majeed et al., 2015*; *Hassan et al., 2021*). *Meena et al. (2015)* isolated PGPR from peas surviving at 5 °C.

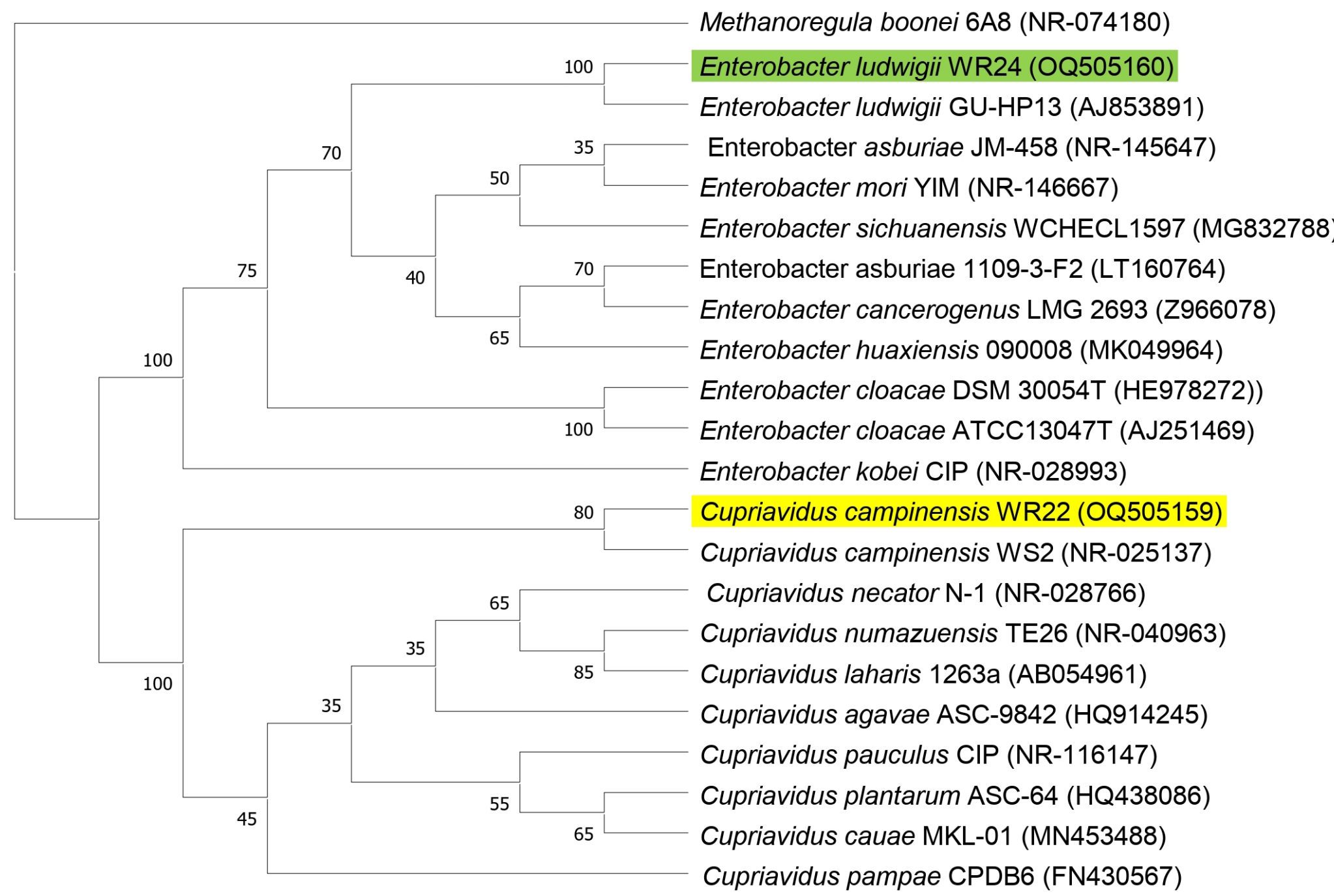

**Figure 5 Phylogenetic tree based on 16S rRNA gene sequencing of potential bacteria using neighbor-joining method.** WR24 was positioned in clade 1 and WR22 in clade 2. Genbank accession numbers are written in paranthesis. The values at the nodes are bootstrap values which indicates the level of confidence on branch position. *Methanoregula boonei* 6A8 (NR-074180), *Enterobacter ludwigii*_WR24 (OQ505160), *Enterobacter ludwigii* GU-HP13 (AJ853891), *Enterobacter asburiae* JM-458 (NR-145647), *Enterobacter mori* YIM (NR-146667), *Enterobacter sichuanensis* WCHECL1597 (MG832788), *Enterobacter asburiae* 1109-3-F2 (LT160764), *Enterobacter cancerogenus* MG 2693 (Z96078), *Enterobacter huaxiensis* 090008 (MK049964), *Enterobacter cloacae* DSM 30054T (HE978272), *Enterobacter cloacae* ATCC13047T (AJ251469), *Enterobacter kobei* CIP (NR-028993), *Cupriavidus campinensis*_WR22 (OQ505159), *Cupriavidus campinensis* WS2 (NR-025137), *Cupriavidus necator* N-1 (NR-028766), *Cupriavidus numazuensis* TE26 (NR-040963), *Cupriavidus laharis* 1263a (AB054961), *Cupriavidus agave* ASC-9842 (HQ914245), *Cupriavidus pauculus* CIP (NR-116147), *Cupriavidus plantarum* ASC-64 (HQ438086), *Cupriavidus cauae* MKL-01 (MN453488), *Cupriavidus pampae* CPDB6 (FN430567).

After the cold-tolerance assay, the bacteria were characterized for biochemical assays and plant growth promotion ability. PGPR have evolved direct and indirect mechanisms to increase plant growth (*Glick, 1995*). Direct mechanisms are involved in regulating the growth hormones and macronutrients that affect plant growth directly. This include phosphate solubilization, IAA production, nitrogen fixation, *etc*. Indirect mechanisms are generally happening outside the plant and bacteria increase plant growth indirectly by killing plant pathogens and increasing resistance against abiotic stress (*Jha et al., 2011*; *Glick, 2014*; *Ramos-Solano, Barriuso & Gutiérrez-Ma nero, 2008*).

Phosphate is the second most important nutrient after nitrogen for plant growth. It helps in root hair development and improves plant growth (*Ma et al., 2001*). In this study, WR24 showed maximum phosphate solubilization at 4 °C, while WR15 showed maximum phosphate solubilization at 28 °C. Previously, *Subramanian et al. (2016)* isolated cold-tolerant phosphate solubilizing bacteria from agricultural soil and described their potential

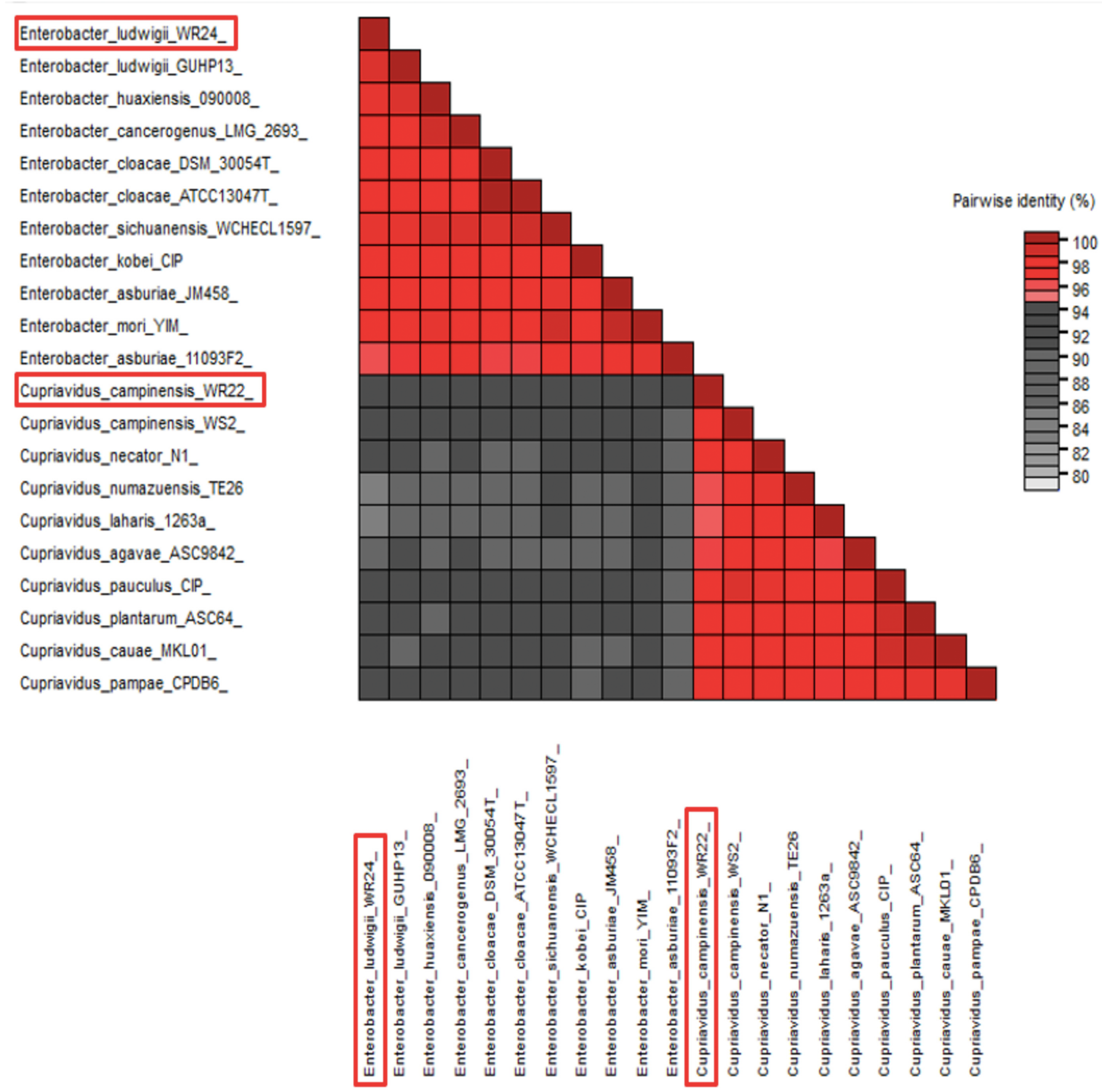

**Figure 6 Pairwise identity chart based on 16S rRNA gene sequences of potential bacteria using SDT.** WR24 showed 99.9% similarity with *Enterobacter ludwigii* and WR22 showed 99.78% similarity with *Cupriavidus campinesis*.

to increase tomato plant growth. Indole-3-acetic acid plays a very important role in root growth and development, which helps in the uptake of nutrients (*Sabkia et al., 2021*). In this study, it was found that the production of IAA was reduced at low temperatures as compared to mesophilic conditions. WR24 showed the highest production of IAA at 4 °C and 28 °C whereas WR24 showed a higher production of IAA at 28 °C. Similar results are demonstrated by *Mishra et al. (2011)* and *Mantelin et al. (2006)*. Nitrogen is the most important nutrient required for plant growth and a major constitute of genetic material. In this study, WR22 showed maximum nitrogen fixation ability at 4 °C, while WR18 exhibited the highest ability at 28 °C. The amount of nitrogen fixation was higher at 4 °C as compared to 28 °C. Our results are in agreement with *Chumpitaz-Segovia et al. (2020)*, who reported efficient nitrogen fixation of the *Chenopodium quinoa*-associated bacteria at 6 °C and 25 °C. Cellulase and pectinase are involved in the decomposition of dead organic matter present in the soil and enrich the soil nutrient profile (*Reetha et al., 2014*). In this study, maximum cellulase production at 4 °C was shown by WR22 while at 28 °C by WR20. Maximum pectinase production was observed by WR22 at both 4 °C and 28 °C, but the

amount of pectinase was higher at 4 °C. *Yadav et al. (2016)* also explained that cold-tolerant *Bacillus* isolated from glacial lakes has the potential to produce cellulase and pectinase at 4 °C, which help to break cell wall of phytopathogens. Zinc is a micronutrient that helps in the synthesis of proteins, and nucleic acid metabolism and acts as a cofactor for auxin production (*Tsonev & Cebola Lidon, 2012*). In this study, WR24 and WR23 showed the highest zinc solubilization efficiency at 4 °C while at 28 °C the maximum efficiency was shown by WR19 and WR20 with ZnO and $ZnCO_3$, respectively. *Costerousse et al. (2018)* also isolated bacteria from wheat rhizosphere and reported zinc solubilization efficiency at 28 °C by using ZnO as a zinc source.

Biofilm formation is a major characteristic of bacteria to survive under stress conditions. In this study, WR24 showed the highest biofilm formation ability at both 4 °C and 28 °C. The amount of biofilm was higher at 4 °C as compared to 28 °C. Our results are in agreement with *Zubair et al. (2019)* who reported the biofilm formation potential of psychrotolerant *Bacillus* strains isolated from Tibetan prefectures at 4 °C and 28 °C.

Over all, *in vitro* testing revealed that each bacterium has an optimum temperature to perform its inherent character. Some bacteria perform a specific character at low temperatures while other bacteria do at high temperature (*Junge, Cameron & Nunn, 2019*). WR22 and WR24 showed better plant growth-promoting characters at 4 °C because these bacteria are adapted to cold conditions and work efficiently at low temperatures. Overall production of plant growth-promoting characters was high at 28 °C as compared to 4 °C, which might be due to the reduction in bacterial metabolic activities at 4 °C. *Zeng, Wu & Wen (2016)* and *Vega-Celedon et al. (2021)* also explained that plant growth-promoting efficiency is reduced at a lower temperature.

Phylogenetic analysis of potential bacteria revealed the maximum similarity of WR22 with *Cupriavidus campinesis* and WR24 with *Enterobacter ludwigii*. This is the first report on the novel association of *C. campinesis* and *E. ludwigii* in the wheat rhizosphere. Previously, *Platero et al. (2016)* reported *Cupriavidus campinesis* as a novel species isolated from root nodules of the Mimosa plant. Isolation of *E. ludwigii* from the rhizosphere of *Lolium perenne* and rice was reported in many studies (*Shoebitz et al., 2009*; *Lee et al., 2019*). *Dayamrita & Paul (2021)* also isolated *E. ludwigii* from the roots of the medicinal plant *Leucas aspera*.

Inoculation of *Cupriavidus campinesis* WR22 and *Enterobacter ludwigii* WR24 exhibited the maximum potential to enhance plant growth parameters under controlled conditions. Previously, *Singh et al. (2018)* revealed that inoculation of *E. ludwigii* can increase wheat growth parameters under controlled conditions experiment. *Adhikari et al. (2020)* inoculated rice with *E. ludwigii*, which increased the plant growth parameter under cadmium stress. *Zaballa, Golluscio & Ribaudo (2020)* reported that *E. ludwigii* improved the growth of barley up to 56%. *Kang et al. (2021)* also demonstrated that the inoculation of *E. ludwigii* increased the growth of alfalfa under cold-stress conditions. Recently, *Zheng et al. (2023)* reported that inoculation of *Cupriavidus sp.* increases plant growth parameters in *Brassica napus* plants. Taking all the findings together, it is suggested that *C. campinesis* WR22 and *E. ludwigii* WR24 have maximum PGP potential and could be used as cold-tolerant biofertilizers after field trials.

## CONCLUSIONS

Psychrotolerant PGPR have the potential to increase the growth of wheat in cold areas. In this study, ten bacteria were isolated and five efficient bacteria were subjected to plant assay based on the cold-tolerance and plant growth-promoting properties. Inoculations of WR22 and WR24 were found promising for wheat growth promotion. Phylogenetic analysis revealed the highest similarity of WR22 with *Cupriavidus campinesis* and WR24 with *Enterobacter ludwigii*. This is the first report on the novel association of *C. campinesis* WR22 and *E. ludwigii* WR24 in the wheat rhizosphere. Conclusively, *C. campinesis* WR22 and *E. ludwigii* WR24 could be used as cold-tolerant biofertilizers after field trials.

## ACKNOWLEDGEMENTS

The authors are thankful to Dr. Asma Imran, Principal Scientist, Environmental Biotechnology Division, National Institute for Biotechnology and Genetic Engineering (NIBGE), Pakistan, for improving the write-up of manuscript.

### Funding

The authors received no funding for this work.

### Competing Interests

Mohsin Tariq is an Academic Editor for PeerJ.

### Author Contributions

- Muhammad Abdullah conceived and designed the experiments, performed the experiments, analyzed the data, prepared figures and/or tables, and approved the final draft.
- Mohsin Tariq conceived and designed the experiments, analyzed the data, prepared figures and/or tables, authored or reviewed drafts of the article, and approved the final draft.
- Syeda Tahseen Zahra performed the experiments, analyzed the data, prepared figures and/or tables, authored or reviewed drafts of the article, and approved the final draft.
- Azka Ahmad conceived and designed the experiments, performed the experiments, prepared figures and/or tables, and approved the final draft.
- Marriam Zafar analyzed the data, authored or reviewed drafts of the article, and approved the final draft.
- Shad Ali analyzed the data, authored or reviewed drafts of the article, and approved the final draft.

### DNA Deposition

The following information was supplied regarding the deposition of DNA sequences:

The data are available at GenBank: OQ505159 for WR22 and OQ505160.

## Data Availability

The raw data are available in the Supplementary File.

## Supplemental Information

Supplemental information for this article can be found online at http://dx.doi.org/10.7717/peerj.16399#supplemental-information.

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
