# Peer review of "Potential of psychrotolerant rhizobacteria for the growth promotion of wheat (Triticum aestivum L.)"

_PeerJ, doi:10.7717/peerj.16399_

## Round 0.1 · original submission · Major Revisions

Kindly respond to all the comments and queries raised by the reviewers in a rebuttal letter.

·

Basic reporting

This study isolated 10 strains of psychrotolerant bacteria from the rhizosphere of wheat at 4 ℃ and tested their cold resistance and growth promoting properties. Plant experiments were conducted on 5 effective bacteria, and it was found that the Cupriavidus campensis WR22 and Enterobacter ludwigii WR24 strains can be used as cold resistant biological fertilizers. This study has certain value for increasing wheat production in Pakistan. However, based on the current version, there still exists some limitations in the editing of manuscript.
1. Line 92-100 This part of the introduction about cold stress on plant transcriptional changes is not very consistent with the content of this study, because this study did not cover the content of plant transcriptional changes.
2. The introduction should be extended to discuss the hypothesis and research question in detail.

Experimental design

1. Why is King’s B media the only method used to isolate psychrotolerant bacteria? This medium cannot be used to isolate most common bacteria.
2. Lack of data analysis in Materials and Methods
3. The sequence of the Methods and the Results corresponds, with the Phylogenetic analysis of potential bacteria being the last part of the Methods and the Controlled conditions experiment being the last part of the Results.
4. It is important to provide some pictures of wheat growth after inoculation to better illustrate the growth promoting effect of the strain on the wheat.

Validity of the findings

1. In general, the Discussion is not well-focused; One paragraph discussion of each PGP feature is not a good format; Reporting data from other papers without trying to give an elaborate analysis. In discussion, do not make statements that are not supported by your data. For example, line370-371, 45℃ isolated bacteria from wheat rhizosphere.

Additional comments

1. PRPG was studied, and PGPB came up in the discussion, the two are not strictly the same meaning.

Reviewer 2 ·

Basic reporting

Here, the authors describe ten psychrotolerant rhizospheric bacteria isolated from wheat, which have different features regarding phosphorus solubilization, indole-3-acetic acid production, nitrogen fixation, cellulase production, pectinase production, zinc mobilization and biofilm formation, as well as to induce the wheat growth. Also, phylogenetic analysis showed that two of those bacteria correspond to Cupriavidus campinensis (W22) and Enterobacter ludwigii (W24). In general, the work is well-written, and the results are very interesting. However, there are some points that they have to improve:

You have many typing errors, for example:
Line 154: Please correct “temperature”
Line 209: Please correct “Pectinase activity of …”
Line 233: Please correct “completely randomized design”.
Line 234: Do you mean “bacterial pellet”?
Line 239: Please specify at what concentration you used the inoculum
Line 327: Please correct “Sequences were …”.
Line 329: Please specify which other organisms you used to construct the tree.
Line 371: Please correct “recently”

Figure 4: Please indicate what the numbers of each branch mean.
-Complementing Table 1 with photographs to appreciate the bacterial morphology is highly desirable.
Table 2: You need to complement this information with photographs to compare your results. Also, in the description, you need to indicate which statistical analysis you used and the p-value.

-I’m concerned about the number of samples that you used. Usually, n = 3 it’s not sufficient to evaluate the effect of the bacteria on plants. Also, you didn’t mention if you repeat these experiments at least twice. Please clarify.
-To enrich your discussion, I suggest giving more details on the mechanisms that the bacteria use to induce all the plant growth-promoting characteristics.
-To discuss your phylogenetic analysis, please detail more about the clades where your bacteria were found. What characteristics do these clades have? Do these clades follow a specific classification? evolutionarily speaking, what did you find?

Experimental design

The material and methods are well described.

Validity of the findings

The statistical analysis of all plant growth-promoting characteristics is missing. Please include them.

Additional comments

NA

---

## Round 0.2 · accepted · Accept

Thanks for addressing the revisions requested. Now, your manuscript is accepted in PeerJ.

Please define "PGPR" in the abstract/ conclusion when first mentioned.

·

Basic reporting

It is my great honor to review PeerJ manuscript “Potential of psychrotolerant rhizobacteria for the growth promotion of wheat (Triticum aestivum L.)” again. I am glad that this article has been greatly improved and basically solved the problems I raised. The discussion part of this manuscript adds a lot of content and analyzes the reasons based on the existing results. The picture of wheat growth after inoculation to better illustrate the growth after inoculation was added. The language and description of this manuscript have also been greatly improved. Therefore, I think this manuscript is ready for further publication.

Experimental design

There was only one point, missing the “Statistical analysis” section in the “Materials & Method” section.

Validity of the findings

no comment

Additional comments

no comment

Reviewer 2 ·

Basic reporting

The authors attended to all the comments and suggestions. Now, the manuscript is improved.

Experimental design

No comment

Validity of the findings

No comment

Additional comments

No comment